# New Terpendole Congeners, Inhibitors of Sterol *O*-Acyltransferase, Produced by *Volutella citrinella* BF-0440

**DOI:** 10.3390/molecules25133079

**Published:** 2020-07-06

**Authors:** Elyza Aiman Azizah Nur, Keisuke Kobayashi, Ai Amagai, Taichi Ohshiro, Hiroshi Tomoda

**Affiliations:** 1Department of Microbial Chemistry, Graduate School of Pharmaceutical Sciences, Kitasato University, Tokyo 108-8641, Japan; ml18124@st.kitasato-u.ac.jp (E.A.A.N.); kobayashikei@pharm.kitasato-u.ac.jp (K.K.); tohshiro@med.nagoya-u.ac.jp (T.O.); 2Medicinal Research Laboratories, School of Pharmacy, Kitasato University, Tokyo 108-8641, Japan; 3Department of Microbial Chemistry, School of Pharmacy, Kitasato University, Tokyo 108-8641, Japan; pp14007@st.kitasato-u.ac.jp; 4ITOCHU Collaborative Research-Molecular Targeted Cancer Treatment for Next Generation, Graduate School of Medicine, Nagoya University, Aichi 466-8550, Japan

**Keywords:** terpendole, sterol *O*-acyltransferase, SOAT, inhibitor, fungal metabolite

## Abstract

New terpendoles N-P (**1–3**) were isolated along with 8 structurally related known compounds including terpendoles and voluhemins from a culture broth of the fungus *Volutella citrinella* BF-0440. The structures of **1–3** were elucidated using various spectroscopic experiments including 1D- and 2D-NMR. All compounds **1–3** contained a common indole–diterpene backbone. Compounds **2** and **3** had 7 and 6 consecutive ring systems with an indole ring, respectively, whereas **1** had a unique indolinone plus 4 consecutive ring system. Compounds **2** and **3** inhibited both sterol *O*-acyltransferase 1 and 2 isozymes, but **1** lost the inhibitory activity. Structure–activity relationships of fungal indole–diterpene compounds are discussed.

## 1. Introduction

Natural compounds derived from animals, plants, and microbes have historically been used as a rich source for new drug discovery. These sources provide more structurally diversity than synthetic compounds [1]. Generally, a set of structurally related compounds tend to be produced by a single organism, allowing us to identify and associate the chemical groups responsible for evoking a target biological effect. This structure–activity relationship (SAR) information then becomes a key approach for lead optimization in drug discovery.

Sterol *O*-acyltransferase (SOAT), an enzyme that catalyzes the formation of cholesteryl ester (CE), has been a potential target for development as a post-statin drug. Researchers now understand that SOAT has two isozymes, SOAT1 and SOAT2, with distinct functions in the human body [2,3,4,5], and that selective inhibition of SOAT2 is responsible for the prevention of atherosclerosis and fatty liver disease [6,7,8,9,10]. Accordingly, our group has been focusing on the search for SOAT2 selective inhibitors from microbial resources. As a result, we found that fungal pyripyropene A was a highly selective SOAT2 inhibitor [11]. After this finding, we continued screening microbial culture broths for SOAT2 inhibitors, but the hit rate was very low. Recently, we discovered new indoline–diterpene-containing compounds, termed voluhemins A (**4**) and B (**5**), and NK12838 (**6**) [12], from culture broth of the fungus *Volutella citrinella* BF-0440 [13]. Notably, **5** was found to be a selective SOAT2 inhibitor. Furthermore, the culture broth of the fungus still contained other structurally related compounds. As a result, new terpendoles N-P (**1**–**3**), were isolated from the culture broth along with four known terpendoles (**7**–**10**) [14,15] and tolypocladin A (**11**) [16] and presented in Figure 1. In the present study, we describe the isolation, structural elucidation, and SOAT inhibitory activity of these compounds and discussed the SARs of indole/indoline-diterpenes in SOAT1/SOAT2 inhibitory activity.

## 2. Results

### 2.1. Isolation of New Terpendoles and Related Compounds

The isolation procedure of new terpendoles (**1–3**) and known related compounds (**4–11**) from *V. citrinella* BF-0440 culture broth is summarized in Figure 2. The crude extract (2.5 g) was fractionated by an ODS column with stepwise gradient elution (40% aq CH_3_CN, 60% aq CH_3_CN, 80% aq CH_3_CN, and 100% CH_3_CN solutions, fractionated into two). Voluhemins A (**4**) and B (**5**) were purified from fraction 80%-1, as reported in detail previously [13]. Fraction 60%-2 (249 mg) was subjected to preparative high-performance liquid chromatography (HPLC) to give NK12838 (**6**) [12] and new terpendole N (**1**). Purification from fraction 80%-2 (214 mg) by preparative HPLC allowed isolation of three peaks, new terpendole O (**2**), terpendole C (**7**) [14], and a third peak. The third peak was, however, found to be a mixture of related compounds from proton NMR analyses. This mixture was also obtained from fraction 100%-1 (662 mg) by preparative HPLC along with terpendoles C (**7**) [14], D (**8**) [14], and L (**9**) [15], and tolypocladin A (**10**) [16]. The mixture was finally separated in different HPLC conditions to yield a new terpendole P (**3**) and a known terpendole J (**11**) [15]. All of these peaks were collected and concentrated to dryness to give **1** (2.1 mg), **2** (21.6 mg), **3** (2.7 mg), **6** (46.6 mg), **7** (29.7 mg and 15.7 mg from 80%-2 and 100%-1, respectively), **8** (8.3 mg), **9** (6.8 mg), **10** (3.8 mg) and **11** (1.6 mg) as white powders.

### 2.2. Structural Elucidation of New Terpendole O (***2***)

The physicochemical properties of **2** are summarized in Table 1. The molecular formula C_37_H_49_NO_6_ of **2** was determined based on HR-ESIMS (*m*/*z*, found 604.3638, calcd 604.3625 for C_37_H_50_NO_6_ [M + H]^+^), indicating 14 degrees of unsaturation. The characteristic UV absorbance maxima at 230 nm and 283 nm, along with IR absorptions at 3449, 2979, 2930, and 1458 cm^−1^, were similar to those of reported terpendoles [14,15], suggesting **2** possesses a similar chemical skeleton. The structure of **2** was then elucidated using NMR experiments (Appendix A). The data from ^13^C-NMR and ^1^H-NMR in DMSO-*d*_6_ are shown in Table 2. The ^13^C-NMR spectrum of **2** showed 37 resolved signals, which were classified into 8 methyl carbons, 6 methylene carbons, four *sp*^2^ methine carbons, 7 *sp*^3^ methine carbons, 6 *sp*^2^ quaternary carbons, and 6 *sp*^3^ quaternary carbons using an analysis of heteronuclear single-quantum correlation (HSQC) data (Table 2 and Appendix A). The ^1^H-NMR spectrum of **2** showed 49 proton signals (Table 2 and Appendix A), which were classified into 8 methyl protons, 6 methylene protons, four *sp*^2^, and 7 *sp*^3^ methine protons, one amine proton and one hydroxy proton. The connectivity of proton and carbon atoms was established using HSQC (Table 2). These data are in accordance with the molecular formula. As shown in Figure 3a, analysis of the ^1^H-^1^H correlation spectroscopy (COSY) spectrum gave 6 partial structures I to VI; (I) 5-H_2_ (*δ* 1.67, 2.42), 6-H (*δ* 2.12) and 7-H (*δ* 4.26), (II) 9-H (*δ* 3.40), 10-H (*δ* 4.04) and 11-H (*δ* 3.50), (III) 14-H (*δ* 1.46, 1.52), 15-H_2_ (*δ* 1.46, 1.80), 16-H (*δ* 2.71) and 17-H_2_ (*δ* 2.45, 2.71), (IV) 21-H (*δ* 6.71), 22-H (*δ* 6.86) and 23-H (*δ* 7.12), (V) 31-H (*δ* 5.50), and 33-H (*δ* 5.10), and (VI) 37-H_2_ (*δ* 2.93, 3.01), and 38-H (*δ* 2.93). The COSY correlations of 14-H_2_ (*δ* 1.43, 1.56) and 15-H_2_ (*δ* 1.60, 1.90), and 37-H_2_ (*δ* 2.98, 3.29) and 38-H (*δ* 3.09) were more clearly observed in CDCl_3_ (Appendix A and Appendix A). Furthermore, the following linkages including partial structures I to VI were elucidated using ^13^C-^1^H long range couplings of ^2^*J* and ^3^*J* in the heteronuclear multiple bond correlation (HMBC) spectrum (Figure 3a). (1) The cross peaks from 21-H and 23-H to C-19 (*δ* 124.1) and from 22-H to C-20 (*δ* 128.0) and C-24 (*δ* 139.7) indicated the presence of a trisubstituted benzene ring containing partial structure IV. The cross peaks from 1-NH (*δ* 10.69) to C-2 (*δ* 152.3), C-18 (*δ* 114.3), C-19, and C-24 indicated that a pyrrole ring is directly attached to the benzene, showing the presence of an indole moiety. (2) The cross peaks from 37-H_2_ (*δ* 2.93, 3.01) to C-39 (*δ* 57.9), from 40-H_3_ (*δ* 18.7) to C-38 (*δ* 63.4), C-39, and C-41 (*δ* 24.6) and from 41-H_3_ (*δ* 24.6) to C-38, C-39, and C-40 suggested the presence of an isopentanyl unit containing partial structure VI. Taking the chemical shifts of C-38 and C-39 into consideration, these carbons were involved in the formation of an epoxide moiety. The cross peaks from 21-H to C-37 (*δ* 32.3) and from 38-H to C-20 supported that an epoxy-isopentanyl unit is connected to C-20 of the indole moiety. (3) The cross peaks from 17-H_2_ to C-2, C-3 (*δ* 49.9), and C-18 suggested that a cyclopentene ring (ring A) is attached to the indole ling. (4) The cross peaks from 25-H_3_ (*δ* 1.16) to C-3 and C-4 (*δ* 42.1) and from 26-H_3_ (*δ* 1.02) to C-3, C-4, C-5 (*δ* 25.5), and C-13 (*δ* 76.5) and the clear cross peaks from 5-H_2_ (*δ* 1.34, 2.70) to C-13 (*δ* 78.0), 6-H_2_ (*δ* 2.28, 1.78) to C-4 (*δ* 42.3), and 7-H (*δ* 4.38) to C-12 (*δ* 67.8) in CDCl_3_ supported that two cyclohexane rings (rings B and C) are attached to ring A, covering partial structure I and III. The cross peak from 13-OH (*δ* 4.50) to C-4 indicated the presence of a hydroxy group at quaternary C-13. (5) The cross peaks from 7-H to C-9 (*δ* 71.0) and C-11 (*δ* 58.9), from 11-H to C-7 (*δ* 70.6) and C-12 (*δ* 67.0), from 9-H to C-27 (*δ* 74.1), C-28 (*δ* 16.7), and C-29 (*δ* 28.2), from 10-H to C-27, from 28-H_3_ (*δ* 1.22) to C-9, C-27, and C-29, from 29-H_3_ (*δ* 1.12) to C-9, C-27, and C-28, and from 31-H to C-10 (*δ* 70.1) and C-27 showed the presence of a tetrahydropyran ring (ring D) and 1,3-dioxane ring (ring E), which includes a partial structure II. In addition, the cross peaks from 31-H to C-34 (*δ* 137.4), from 33-H to C-35 (*δ* 18.3) and C-36 (*δ* 25.0), from 35-H_3_ (*δ* 1.63) to C-33 (*δ* 122.5), C-34, and C-36, and from 36-H_3_ (*δ* 1.64) to C-33, C-34, and C-35 suggested that a 2-methyl-2-butene with partial structure V is bound to C-31 of ring E. Taking the molecular formula, the carbon chemical shifts and the degrees of unsaturation into consideration, two oxygen atoms should build two epoxy groups between C-38 and C-39 and between C-11 and C-12. Taken together, the planar structure of **2** was elucidated as shown in Figure 3a.

To elucidate the relative stereochemistry of 11 chiral carbons in **2**, ^1^H-^1^H coupling constants and nuclear Overhauser effect spectroscopy (NOESY) experiments in DMSO-*d*_6_ and CDCl_3_ were analyzed (Figure 3b). The large coupling constant (9.6 Hz) between 9-H and 10-H indicated that they are in an axial position. In DMSO-*d*_6_, nuclear Overhauser effect (NOE) correlations were observed between 25-H_3_ and 5-H_a_ (*δ* 2.42) and 13-OH and 7-H. In CDCl_3_, additional NOEs were observed between 25-H_3_ (*δ* 1.27) and 15-H_a_ (*δ* 1.90), 25-H_3_ and 17-H_a_ (*δ* 2.60), 5-H_a_ (*δ* 2.70) and 6-H_a_ (*δ* 2.28), 6-H_a_ and 7-H (*δ* 4.38), and 7-H and 9-H (*δ* 3.57). These data indicated that these protons were oriented in the same side of the plane of the consecutive ring system. In contrast, NOEs were observed between 16-H and 26-H_3_, 10-H and 26-H_3_, and among 10-H, 31-H, and 28-H_3_ in DMSO-*d*_6_, and between 16-H (*δ* 2.80) and 17-H_b_ (*δ* 2.83), 6-H_b_ (*δ* 1.78) and 26-H_3_ (*δ* 1.14) in CDCl_3_. The data indicated that these protons were present on the opposite side. Furthermore, NOE was observed between 11-H (*δ* 3.61) and 14-H_a_ (*δ* 1.43) in CDCl_3_, suggesting that 11-H was in the equatorial position. Taken together, the relative stereochemistry of **2** except for C-38 was elucidated to be 3*S**, 4*R**, 7*S**, 9*S**, 10*R**, 11*R**, 12*S**, 13*S*,* 16*S**, and 31*S**.

### 2.3. Structural Elucidation of New Terpendole N (***1***)

The physicochemical properties of **1** are summarized in Table 1. The molecular formula C_37_H_51_NO_9_ for **1** was assigned based on HR-ESIMS (*m*/*z*, found 654.3637, calcd 654.3642 for C_37_H_52_NO_9_ [M + H]^+^), indicating 13 degrees of unsaturation. Compound **1** had UV absorbance maxima at 213, 248, and 290 nm, along with IR absorptions at 3413, 2977, 2930, and 1458 cm^−1^. The structure of **1** was then elucidated using NMR experiments (Appendix A). The data of ^13^C-NMR and ^1^H-NMR in DMSO-*d*_6_ are shown in Table 2. The ^13^C-NMR spectrum of **1** showed 37 resolved signals, which were classified into 8 methyl carbons, 6 methylene carbons, four *sp*^2^ methine carbons, 7 *sp*^3^ methine carbons, 5 *sp*^2^ quaternary carbons, and 7 *sp*^3^ quaternary carbons by an analysis of HSQC data (Table 2 and Appendix A). The ^1^H-NMR spectrum of **1** showed 51 proton signals (Table 2 and Appendix A), which were classified into 8 methyl protons, 6 methylene protons, four *sp*^2^ and 7 *sp*^3^ methine protons, one amine proton, and three hydroxy protons. The connectivity of proton and carbon atoms was established by HSQC (Table 2). Compared with the ^13^C-NMR spectra of **2**, the chemical shift values of **1** at C-2, C-3, and C-18 were remarkably shifted, suggesting that **1** has one additional *sp*^2^ and two additional *sp*^3^ oxygenated carbons. From the ^13^C-^1^H long range couplings of ^2^*J* and ^3^*J* in the HMBC spectrum (Figure 4a and Appendix A), the cross peaks from 18-OH (*δ* 5.72) to oxygenated C-18 (*δ* 76.8) and aromatic C-19 (*δ* 128.9), and from 3-OH (*δ* 3.74) to oxygenated C-3 (*δ* 74.6), C-16 (*δ* 37.5), and C-25 (*δ* 19.5) indicated that a hydroxy moiety was directly bound to C-18 and C-3. Furthermore, the chemical shift value at C-2 (*δ* 180.2) supported the presence of an amide group, which implied that **1** contains an indolinone moiety. Taken together, the planar structure of **1** was proposed as shown in Figure 4a,b, which fulfilled the molecular formula and degrees of unsaturation.

Relative stereochemistry of **1** was elucidated by analyses of ^1^H-^1^H coupling constants and NOESY experiments. As shown in Figure 4c, the NOE correlations of **1** were very similar to those of **2**. Consequently, the relative stereochemistry of 12 chiral carbons except for C-18 and C-38 of **1** was determined to be 3*S**, 4*R**, 7*S**, 9*S**, 10*R**, 11*R**, 12*S**, 13*S*,* 16*S**, and 31*S**.

### 2.4. Structural Elucidation of New Terpendole P (***3***)

The physicochemical properties of **3** are summarized in Table 1. The molecular formula C_32_H_41_NO_5_ was determined based on HR-ESIMS (*m*/*z*, found 520.3063, calcd 520.3067 for C_32_H_42_NO_5_ [M + H]^+^). Compound **3** had UV absorbance maxima at 227 and 280 nm, along with IR absorptions at 3380, 3004, 2931, and 1681 cm^-1^ (Table 1). The structure of **3** was then elucidated using NMR experiments (Appendix A). The data of ^13^C-NMR and ^1^H-NMR in DMSO-*d*_6_ are shown in Table 3. The ^13^C-NMR spectrum of **3** showed 32 resolved signals, which were classified into 6 methyl carbons, 5 methylene carbons, 5 *sp*^2^ methine carbons, 6 *sp*^3^ methine carbons, 5 *sp*^2^ and 5 *sp*^3^ quaternary carbons using HSQC experiments (Table 2 and Appendix A). The ^1^H-NMR spectrum of **3** showed 41 proton signals (Table 2 and Appendix A), which were classified into 6 methyl protons, 5 methylene protons, 5 *sp*^2^ and 6 *sp*^3^ methine protons, one amine proton, and one hydroxy proton. The connectivity of proton and carbon atoms was established using HSQC (Table 2). The ^13^C chemical shifts of **3** were similar to these of terpendole J (**11**) [15] (C_32_H_43_NO_5_) except for C-14 (*δ* 54.7). From the ^13^C-^1^H long range couplings of ^2^*J* and ^3^*J* in the HMBC spectrum (Figure 5a and Appendix A), the position of these carbons was confirmed. Considering the different molecular formula between **3** and **10**, the presence of an epoxide moiety was suggested between C-13 and C-14. Therefore, the structure of **3** was elucidated as shown in Figure 5a, which fulfilled the molecular formula and degrees of unsaturation. The relative stereochemistry of 10 chiral carbons of **3** was determined to be 3*S**, 4*R**, 7*S**, 9*S**, 10*R**, 11*R**, 12*S**, 13*R** 14*S**, and 16*R** by NOE experiments in a similar way for **1** and **2** (Figure 5b).

### 2.5. Structural Identification of Known Compounds ***4***–***11***

Based on spectral data including ^1^H-NMR, ^13^C-NMR, and MS, and the search results of SciFinder Scholar, the structures of **4****–11** were identified as the known NK12838 (**4**), voluhemins A (**5**) and B (**6**), terpendoles C (**7**), D (**8**), L (**9**), and J (**11**), and tolypocladin A (**10**) [12,14,15,16].

### 2.6. Inhibition of SOAT Isozymes Using SOAT1- and SOAT2-CHO Cells

The human SOAT inhibitory activities of **1**–**11** were evaluated in accordance with our established method in cell-based assays using SOAT1/SOAT2-CHO cells [11,17]. As shown in Table 3, **2** and **3** containing an indole moiety inhibited both SOAT1 and SOAT2 activity to a similar extent (IC_50_ values ranging from 2.4 to 6.9 µM), giving selectivity index (SI; log (IC_50_ for SOAT1/IC_50_ for SOAT2)) values of +0.06 and -0.06, respectively. Thus, we defined that **2** and **3** are dual-type SOAT inhibitors (Table 3) [13]. On the other hand, **1** containing an indolinone moiety showed no inhibitory activity against both SOAT isozymes (IC_50_s, >14 μM). Compounds **1**–**3** did not exert cytotoxic effects on these CHO cell lines even at 14 µM.

## 3. Discussion

During our screening study for SOAT inhibitors, we have reported a series of fungal indole/indoline-containing di/sesquiterpenes; namely terpendoles produced by *Albophoma yamanashiensis* [14,15], sespendole (**12**, Figure 1) produced by *Pseudobotrytis terrestris* FKA-25 [18,19], and voluhemins by *Volutella citrinella* BF-0440 [13]. As described in the present study, a total of 11 terpendole congeners (**1–11**) were isolated from the voluhemin-producing fungus. Herein, the SOAT inhibitory activity (IC_50_ value) and selectivity (SI value) toward SOAT1 and SOAT2 isozymes of these compounds in cell-based assays are summarized in Table 3. Those of **12** are also added to Table 3.

The most common structure from **1–12** and diverse moieties R1 to R6 are illustrated in Figure 6 to discuss the SARs. (1) Ring systems: 5 (**12**), 6 (**3**, **8,** and **11**), and 7 (**2**, **4**, **5**, **6**, **7,** and **9**) consecutive rings including an indole/indoline maintain SOAT inhibitory activity. Compound **1**, which has opened ring A, lost SOAT inhibitory activity. These data indicated that 5–7 consecutive ring systems are important for exhibiting SOAT inhibitory activity. (2) R1 and R2; comparison among compounds **2**, **4**, **5**, **6**, **7**, and **9**; compounds **3**, **8,** and **11**; and **12** indicated that the presence of an isoprenyl-derived moiety at R1 and R2 is not essential for SOAT inhibition. (3) R3 and R4; compounds with an indoline containing a hydroxyl group at R3 and R4 (**4** and **6**) maintained SOAT inhibitory activity, indicating that both hydroxy indoline and indole-type compounds exhibit SOAT inhibition. (4) R5; comparison between **4** and **6** revealed that the presence of a proton at R5 enhances SOAT inhibitory activities (both SOAT1 and SOAT2) approximately 10-fold. (5) R6; comparison between **8** and **11** showed that R6 markedly prefers a proton (**8**) to a hydroxyl group (**11**) for more potent SOAT inhibition (both SOAT1 and SOAT2, approximately 30- to 40-fold). Compound **3** with an epoxy group between C-13 and C-14 partially covering R6 showed an intermediate inhibitory activity. (6) R7; comparison between compounds (**3**, **8,** and **11**) and **10** strongly suggested that the presence of a free hydroxyl group at R7 loses SOAT inhibitory activity. 

Importantly, most terpendole congeners are intrinsically dual-type SOAT inhibitors, whereas only voluhemin B (**5**) exhibited SOAT2 selective inhibition. This finding indicated that the presence of a methoxy moiety at R4 is important for SOAT2 selectivity.

The structure of SOAT localized in the endoplasmic reticulum membrane had not been defined. Very recently, three research groups have just reported the structure of human-SOAT1 using cryo-electron microscopy [20,21,22]. Because SOAT1 and SOAT2 share extensive sequence homology, the structural properties of SOAT2 will be also elucidated in the near future. Accordingly, we will explain the SAR on a molecular basis and the selectivity of SOAT inhibitors (dual-type, SOAT1-selective or SOAT2-selective ones) toward SOAT isozymes.

## 4. Materials and Methods

### 4.1. General

Various NMR spectra were obtained using the NMR System 400 MHz spectrometer (Agilent Technologies, Santa Clara, CA, USA) and 600 MHz spectrometer (Bruker, Karlsruhe, Germany). MS analysis was performed using the AccuTOF LC-plus JMS-T100LP system (JEOL, Tokyo, Japan). Optical rotations were measured using a digital polarimeter (DIP-1000; JASCO, Tokyo, Japan). UV spectra were recorded on a spectrophotometer (8453 UV-visible spectrophotometer; Agilent Technologies). IR spectra were recorded on a Fourier transform IR spectrometer (FT-710; Horiba Ltd., Kyoto, Japan).

### 4.2. Materials

[1-^14^C]Oleic acid (1.85 GBq mmol^−1^) was purchased from PerkinElmer (Waltham, MA, USA). Fetal bovine serum was purchased from Biowest (Nuaille, France). Geneticin (G-418 sulfate) was purchased from Merck (Burlington, MA, USA). MEM vitamin and penicillin (10,000 units mL^−1^)/streptomycin (10,000 mg mL^−1^) solution were purchased from Invitrogen (Carlsbad, CA, USA). Ham’s F-12 medium was purchased from Nacalai Tesque (Kyoto, Japan). Plastic microplates (48-well) were purchased from Corning (Corning, NY, USA).

### 4.3. Fermentation and Isolation

The fermentation procedure for the BF-0440 strain was performed based on a previous report [13]. A loop-full of strain was inoculated and then cultured with shaking for 3 days at 27 °C in 100 mL of seed medium (2.0% glucose, 0.20% yeast extract, 0.050% MgSO_4_·7H_2_O, 0.50% polypeptone, 0.10% KH_2_PO_4_ and 0.10% agar; pH 6.0) in a 500 mL Erlenmeyer flask to obtain a seed culture. The seed culture (10 mL) was inoculated into a 1000 mL culture box containing 50 g of production medium (50 g of brown rice, 5 mL of the following solution; 2.4% PDB, 0.50% MgSO_4_·7H_2_O, 0.50% K_2_HPO_4_ and 0.50% Mg_3_(PO_4_)_2_·8H_2_O). Fermentation was carried out at 27 °C for 14 days in static conditions.

Culture broth was extracted with 70% EtOH (2.5 L). After concentration to remove EtOH, the aqueous solution was extracted with EtOAc (900 mL). The organic layer was collected and EtOAc was evaporated to give crude extract (brown material, 2.5 g). This crude extract was applied to an ODS column (114 g, i.d. 45 × 200 mm) and eluted stepwise with 40% aq CH_3_CN, 60% aq CH_3_CN, 80% aq CH_3_CN, and 100% CH_3_CN solutions (600 mL each, fractionated into two). All fractions were concentrated *in vacuo* and the remaining water layers were extracted with EtOAc. The EtOAc layers of these fractions were collected and evaporated to dryness. Voluhemins A (**4**) and B (**5**) were purified from fraction 80%-1, as reported in detail previously [13]. Fraction 60%-2 (brown material, 249 mg) was subjected to preparative high-performance liquid chromatography (HPLC; column, Pegasil ODS SP100 i.d. 20 × 250 mm; mobile phase, 60% aq CH_3_CN isocratic; flow rate, 6.0 mL min^−1^; detection, UV at 210 nm). In these conditions, NK12838 (**6**) [12] and terpendole N (**1**) were eluted as peaks with retention times of 27 and 29 min, respectively. Purification from fraction 80%-2 (brown material, 214 mg) by preparative HPLC (column, Pegasil ODS SP100 i.d. 20 × 250 mm; mobile phase, 78% aq CH_3_CN isocratic; flow rate, 6.0 mL min^−1^; detection, UV at 210 nm) allowed isolation of three peaks, terpendole O (**2**) eluted at 23 min, terpendole C (**7**) [14] at 24 min, and a third peak at 26 min. The third peak was, however, found to be a mixture of related compounds from proton NMR analyses. This mixture was also obtained from fraction 100%-1 along with terpendoles C (**7**) [14], D (**8**) [14], and L (**9**) [15], and tolypocladin A (**10**) [16]. By preparative HPLC (column, Pegasil ODS SP100 i.d. 20 × 250 mm; mobile phase, 35-min linear gradient 90–95% aq CH_3_CN; flow rate, 6.0 mL min^−1^; detection, UV at 210 nm), **10**, **7**, the mixture, **8,** and **9** were eluted at 16, 18, 22, 31, and 35 min, respectively. The mixture was finally separated in different HPLC conditions (column, Develosil C30 i.d. 20 × 250 mm; mobile phase, 92% aq CH_3_CN isocratic; flow rate, 6.0 mL min^−1^; detection, UV at 210 nm) to yield a new terpendole P (**3**) and a known terpendole J (**11**) [15] eluted as peaks at 22 and 23 min, respectively. All of these peaks were collected and concentrated to dryness to give **1** (2.1 mg), **2** (21.6 mg), **3** (2.7 mg), **6** (46.6 mg), **7** (29.7 mg and 15.7 mg from 80%-2 and 100%-1, respectively), **8** (8.3 mg), **9** (6.8 mg), **10** (3.8 mg) and **11** (1.6 mg) as white powders.

### 4.4. Cell Culture

CHO cells (AC29 cells, SOAT-deficient cells) expressing the SOAT1 or SOAT2 genes from human were cultured using a previously described method [17].

### 4.5. Assay for SOAT Activity in SOAT1- and SOAT2-CHO Cells

Assays for human SOAT1 and SOAT2 activities using SOAT1- and SOAT2-CHO cells were performed using our established method [11,17]. Briefly, SOAT1- or SOAT2-CHO cells (1.25 × 10^5^ cells in 250 µL of medium) were cultured in a 48-well plastic microplate and allowed to recover at 37 °C overnight in 5% CO_2_. At least 80% confluent cells were used for assays. Following overnight recovery, a test sample (in 2.5 µL methanol) and [1-^14^C]oleic acid (in 5 µL 10% EtOH/PBS, 1 nmol, 1.85 kBq) were added to each culture. After a 6-h incubation at 37 °C in 5% CO_2_, medium was removed, and the cells in each well were washed twice with PBS. Cells were lysed by adding 0.25 mL of 10 mM Tris-HCl (pH 7.5) containing 0.1% (*w/v*) sodium dodecyl sulfate, and [^14^C]CE was analyzed using a FLA-7000 analyzer (Fuji Film). In this cell-based assay, [^14^C]CE was produced by the reaction of SOAT1 or SOAT2. SOAT inhibitory activity (%) is defined as ([1-^14^C]CE-drug/[^14^C]CE-control) × 100. The IC_50_ value is defined as the drug concentration causing a 50% inhibition of biological activity.

## Figures and Tables

**Figure 1 molecules-25-03079-f001:**
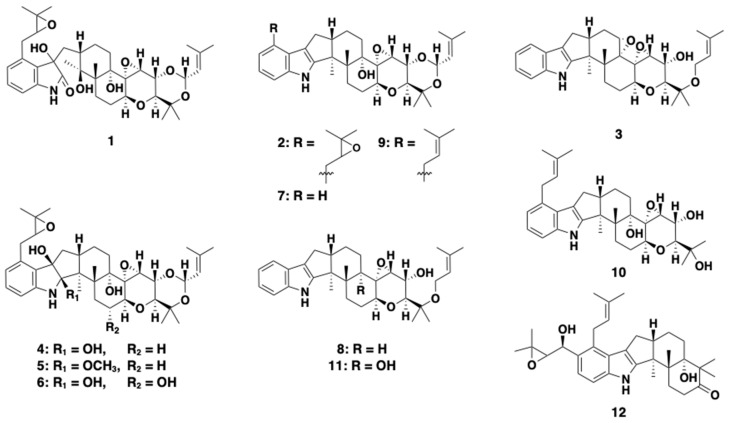
Structures of terpendoles and other known compounds: Terpendole N (**1**), terpendole O (**2**), terpendole P (**3**), voluhemin A (**4**), voluhemin B (**5**), NK12838 (**6**), terpendole C (**7**), terpendole D (**8**), terpendole L (**9**), tolypocladin A (**10**), terpendole J (**11**), and sespendole (**12**).

**Figure 2 molecules-25-03079-f002:**
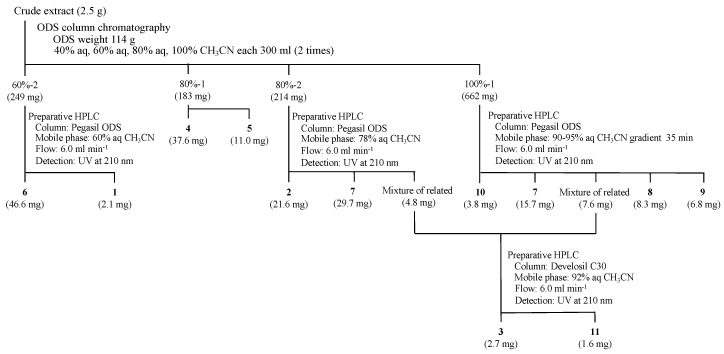
Isolation procedure of terpendoles.

**Figure 3 molecules-25-03079-f003:**
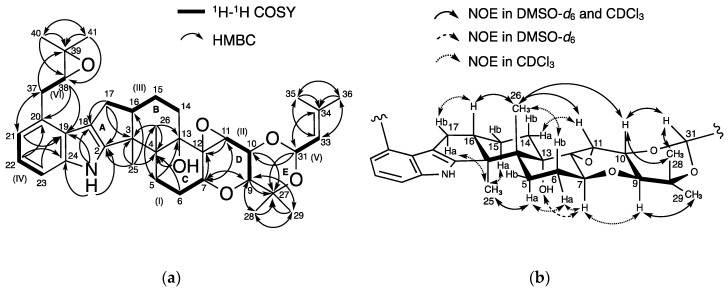
Structural elucidation of **2**: (**a**) Key correlations in ^1^H-^1^H correlation spectroscopy (COSY) and heteronuclear multiple bond correlation (HMBC) spectra; (**b**) nuclear Overhauser effect spectroscopy (NOESY) experiments.

**Figure 4 molecules-25-03079-f004:**
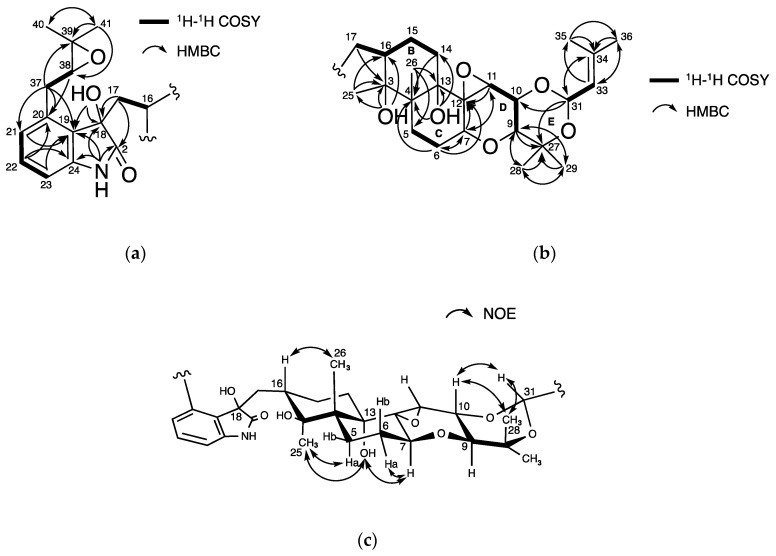
Structural elucidation of **1**: (**a**,**b**) Key correlations in ^1^H-^1^H COSY and HMBC spectra; (**c**) NOESY experiments.

**Figure 5 molecules-25-03079-f005:**
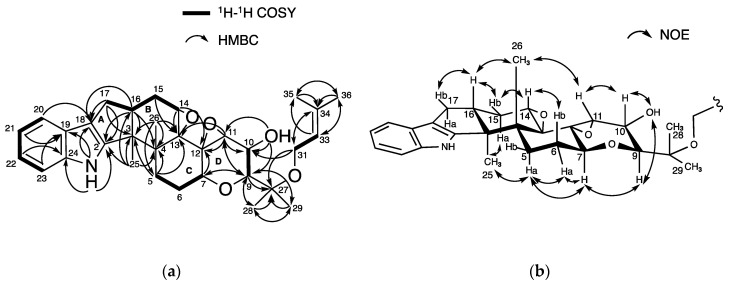
Structural elucidation of **3**: (**a**) Key correlations in ^1^H-^1^H COSY and HMBC spectra; (**b**) NOESY experiments.

**Figure 6 molecules-25-03079-f006:**
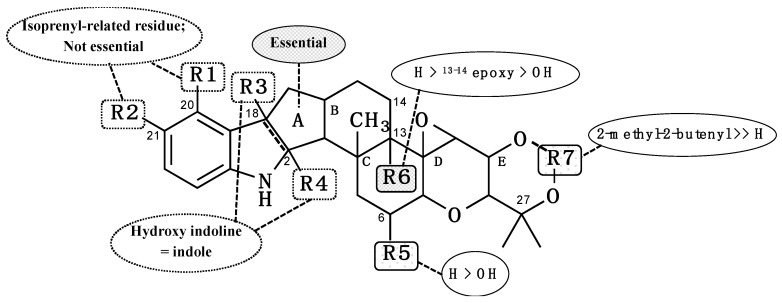
Structure–activity relationships of terpendole congeners for SOAT inhibitory activity.

**Table 1 molecules-25-03079-t001:** Physicochemical properties of **1–3**.

	1	2	3
Appearance	White powder	White powder	White powder
Molecular weight	653	603	519
Molecular formula	C_37_H_51_NO_9_	C_37_H_49_NO_6_	C_32_H_41_NO_5_
HR-ESI-MS (*m/z*)			
Calcd	654.3642 [M + H]^+^	604.3625 [M + H]^+^	520.3067 [M + H]^+^
Found	654.3637 [M + H]^+^	604.3638 [M + H]^+^	520.3063 [M + H]^+^
${{\mathrm{UV}}_{\lambda }}_{\max}^{{\mathrm{CH}}_{3}\mathrm{OH}}\mathrm{nm}\;\left(\log\varepsilon \right)$	213 (4.2), 248 (4.3), 290(3.7)	230 (3.5), 283 (3.0)	227 (4.4), 280 (3.7)
${\mathrm{IR}}_{\max}^{\mathrm{KBr}}\;{\mathrm{cm}}^{-1}$	3413, 2977, 2930, 1718	3449, 2979, 2930, 1458	3380, 3004, 2931, 1681
${\left[\alpha \right]}_{D}^{t}\;\left(c=0.1{\;\mathrm{CH}}_{3}\mathrm{OH}\right)$	−16.6 (t = 25)	−4.0 (t = 23)	−27.8 (t = 25)

**Table 2 molecules-25-03079-t002:** ^1^H- and ^13^C-NMR chemical shifts of **1–3** in DMSO-*d*_6_.

**Terpendole N (1) ^a^**							
**Position**	***δ*_C_^c^, type**	***δ*_H_^d^ (multi, *J* Hz)**	**HMBC**	**Position**	***δ*_C_^c^, type**	***δ*_H_^d^ (multi, *J* Hz)**	**HMBC**
1-NH	-	10.08 (s)	2, 18, 19, 24	18	76.8, C	-	-
2	180.2, C	-	-	18-OH	-	5.72 (s)	18, 19
3	74.6, C	-	-	19	128.9, C	-	-
3-OH	-	3.74 (s)	3, 16, 25	20	136.7, C	-	-
4	44.9, C	-	-	21	121.9, CH	6.81 (d, 7.6)	19, 23
5	23.2, CH_2_	1.15–1.21 (m)	4	22	128.8, CH	7.11 (t, 7.6)	20, 24
		1.92 (m)		23	107.5, CH	6.63 (d, 7.6)	19, 21
6	28.1, CH_2_	1.32 (m)	4, 7, 12	24	141.7, C	-	-
		1.88 (m)		25	19.5, CH_3_	1.08 (s)	3, 16
7	70.5, CH	4.02 (dd, 10.2, 7.4)	6, 11, 12	26	16.7, CH_3_	0.59 (s)	4, 5, 13
				27	74, C	-	-
9	70.9, CH	3.34 (d, 9.6)	27, 28, 29	28	16.7, CH_3_	1.14 (s)	9, 27, 29
				29	28.3, CH_3_	1.073 (s)	9, 27, 28
10	70.1, CH	3.92 (d, 9.6)	-	31	92, CH	5.42 (d, 6.4)	10, 27, 34
11	59.1, CH	3.33 (br s)	7, 9, 12	33	122.5, CH	5.05 (br d, 6.4)	35, 36
12	66.8, C	-	-	34	137.5, C	-	-
13	75.9, C	-	-	35	18.4, CH_3_	1.61 (d, 1.2)	34, 36
13-OH	-	4.18 (s)	4, 13, 14	36	25, CH_3_	1.63 (s)	34, 35
14	25.5, CH_2_	0.81–0.90 (m)	4	37	29.7, CH_2_	2.87 (m)	19, 20, 21, 39
		1.04–1.08 (m)				3.03 (m)	
15	23.7, CH_2_	1.10 (d, 4.8)	17	38	62.9, CH	3.02 (m)	20
		1.27 (s)		39	57.8, C	-	-
16	37.5, CH	1.27–1.34 (m)	-	40	18.9, CH_3_	1.27 (s)	41
17	37.1, CH_2_	1.68 (dd, 12.6, 10.6)	2, 3, 15, 18	41	24.7, CH_3_	1.23 (s)	38, 39, 40
		2.29 (d, 12.6)					
**Terpendole O (2) ^a^**							
**Position**	***δ*_C_^c^, type**	***δ*_H_^d^ (multi, *J* Hz)**	**HMBC**	**Position**	***δ*_C_^c^, type**	***δ*_H_^d^ (multi, *J* Hz)**	**HMBC**
1-NH	-	10.69 (s)	2, 18, 19, 24	18	114.3, C	-	-
2	152.3, C	-	-	18-OH	-	-	-
3	49.9, C	-	-	19	124.1, C	-	-
3-OH	-	-	-	20	128.0, C	-	-
4	42.1, C	-	-	21	118.2, CH	6.71 (d, 7.2)	19, 23
5	25.5, CH_2_	1.67 (br d, 6.4)	26	22	119.5, CH	6.86 (t, 7.6)	20, 24
		2.42 (br d, 6.4)		23	110.1, CH	7.12 (d, 8.0)	19, 21
6	28.53, CH_2_	2.12 (br m)	-	24	139.7, C	-	-
				25	16, CH_3_	1.16 (s)	2, 3, 4
7	70.6, CH	4.26 (t, 8.8)	9, 11	26	18, CH_3_	1.02 (s)	3, 4, 5, 13
				27	74.1, C	-	-
9	71, CH	3.40 (d, 9.6)	7, 27, 28, 29	28	16.7, CH_3_	1.22 (s)	9, 27, 29
				29	28.2, CH_3_	1.12 (s)	9, 27, 28
10	70.1, CH	4.04 (d, 9.6)	27	31	91.9, CH	5.50 (d, 6.4)	10, 27, 34
11	58.9, CH	3.50 (br. s)	7, 9, 12	33	122.5, CH	5.10 (d, 6.4)	35, 36
12	67.0, C	-	-	34	137.4, C	-	-
13	76.5, C	-	-	35	18.3, CH_3_	1.63 (d, 0.8)	33, 34, 36
13-OH	-	4.50 (s)	4	36	25.0, CH_3_	1.64 (s)	33, 34, 35
14	28.38, CH_2_	1.46 (br m)	-	37	32.3, CH_2_	2.93 (m)	21
		1.52 (br s)				3.01 (dd, 16, 8.4)	
15	20.3, CH_2_	1.46 (br m)	-	38	63.4, CH	2.93 (m)	20, 21
		1.80 (br m)		39	57.9, C	-	-
16	49.6, CH	2.71 (br m)	-	40	18.7, CH_3_	1.32 (s)	38, 39, 41
17	28.58, CH_2_	2.45 (br s)	2, 3, 18	41	24.6, CH_3_	1.23 (s)	38, 39, 40
		2.71 (br m)					
**Terpendole P (3) ^b^**							
**Position**	***δ*_C_^c^, type**	***δ*_H_^d^ (multi, *J* Hz)**	**HMBC**	**Position**	***δ*_C_^c^, type**	***δ*_H_^d^ (multi, *J* Hz)**	**HMBC**
1-NH	-	10.81 (s)	2, 18, 19, 24	18	114.7, C	-	-
2	150.6, C	-	-	19	124.1, C	-	-
3	38.3, C	-	-	20	117.8, CH	7.26 (d, 7.8)	18, 22
3-OH	-	-	-	21	118.4, CH	6.88 (td, 7.8, 1.2)	19, 23
4	48.3, C	-	-	22	119.5, CH	6.93 (td, 7.8, 1.2)	20, 24
5	27.9, CH_2_	2.05 (m)	3, 4, 7, 13	23	111.7, CH	7.24 (d, 7.8)	19, 21
				24	140.1, C	-	-
6	28.2, CH_2_	1.75 (m)2.30 (m)	-	25	16.2, CH_3_	0.99 (s)	2, 3, 4, 16
				26	18, CH_3_	1.12 (s)	3, 4, 5, 13
7	70.6, CH	3.86 (m)	9, 12	27	76.9, C	-	-
				28	21.4, CH_3_	1.17 (s)	9, 27, 29
9	75.5, CH	3.25 (d, 9.0)	7, 11, 27, 28, 29	29	23.9, CH_3_	1.16 (s)	9, 27, 28
				31	58.1, CH_2_	3.87 (m)	10, 27, 34
10	65.5, CH	3.89 (m)	-			3.91 (m)	
10-OH	-	4.78 (d, 4.2)	9, 10, 11	33	122.1, CH	5.18 (dd, 7.0, 1.2)	35, 36
11	63.1, CH	3.32 (s)	9, 10, 12	34	134.5, C	-	-
12	62.9, C	-	-	35	17.7, CH_3_	1.58 (s)	34, 36
13	66.9, C	-	-	36	25.3, CH_3_	1.65 (s)	33, 34, 35
14	54.7, CH	3.19 (d, 5.4)	12, 13				
15	24.5, CH_2_	2.02 (m)	13, 17				
		1.85 (m)					
16	45, CH	2.63 (m)	2, 3, 4, 18				
17	26.7, CH_2_	2.17 (m)	2, 3, 15, 18				
		2.62 (m)					

^a 13^C (100 MHz) and ^1^H (400 MHz) spectra were taken on an NMR 400 MHz spectrometer (Agilent). ^b^^13^C (150 MHz) and ^1^H (600 MHz) spectra were taken on an NMR 600 MHz spectrometer (Bruker). Chemical shifts are shown with reference to ^c^ DMSO-*d*_6_ as *δ* 39.5, ^d^ DMSO- *d*_6_ as *δ* 2.48. Multiplicity of signals as follows: s = singlet, d = doublets, dd = double doublets, t = triplet, m = multi. Coupling constants (Hz) were determined by the ^1^H-^1^H decoupling experiments.

**Table 3 molecules-25-03079-t003:** Inhibitory activities of **1**–**12** on human sterol *O*-acyltransferase (SOAT)1 and SOAT2 activities in cell-based assays.

Compound.	IC_50_ (µM) ^a^		SI ^b^ (Type) ^c^
	SOAT1	SOAT2	
**1**	>14	>14	–
**2**	2.8	2.4	+0.06 (dual)
**3**	5.9	6.9	−0.06 (dual)
**4 ^d^**	0.67	0.24	+0.45 (dual)
**5 ^d^**	6.12	0.073	+1.92 (SOAT2)
**6 ^d^**	6.7	3.5	+0.27 (dual)
**7**	2.0	2.2	−0.04 (dual)
**8**	0.8	2.1	−0.42 (dual)
**9**	6.8	1.8	+0.57 (dual)
**10**	>18	>18	–
**11**	18.6	10.7	+0.24 (dual)
**12 ^e^**	12	6.5	+0.26 (dual)

^a^ n ≧ 3. ^b^ Selectivity Index = log(IC_50_ for SOAT1/IC_50_ for SOAT2). ^c^ Type of inhibitors (Dual with -1.0 < SI < +1.0, SOAT1 with SI ≦ −1.0, SOAT2 with +1.0 ≦ SI. ^d^ Data from Ref. 13. ^e^ Data from Ref. 18 (African green monkey SOAT1 and SOAT2-CHO cells).

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
