# Peer review of "New Terpendole Congeners, Inhibitors of Sterol O-Acyltransferase, Produced by Volutella citrinella BF-0440"

_molecules, 2020, doi:10.3390/molecules25133079_

Round 1
Reviewer 1 Report
The manuscript describes the isolation of the three new terpendoles N-P along with eight structurally related known compounds from fermentation of the fungus Volutella citrinella BF-0440. The chemical characterization of new metabolites is well done and very carefully described. The many spectroscopic data obtained and reported for the new compounds fully support the proposed structures, including the relative stereochemistry. Most of the isolated compounds were included in a screening for the sterol O-acyltransferase 1 and 2 isozymes (SOAT1 and SOAT2) inhibitory activity and SARs of the whole fungal indole-diterpene compounds are discussed, revealing new and interesting findings.
The manuscript could be accepted for publication on Molecules after the minor revisions described below:
- The paragraph: 2.1. Isolation of the new terpendoles and related compounds should be moved from the RESULTS section to MATERIAL AND METHODS, it contains too much details which should be reported in the experimental section (which instead lack the description of the isolation and purification methods). It should be replaced by a shorter and more concise description of the isolation procedure; in my opinion, even Scheme 1 is superfluous and, moreover, difficult to read.
- It is not clear to me why the authors use the data in DMSO-d6 for the NMR assignments reported in Table when the CDCl3 spectra are much more resolved. Perhaps their choice was motivated by the possibility of comparing the data reported for previously isolated analogues. However, the high overlap of the signals in DMSO-d6 has led to some misassignments. For example, in the case of the spin system 37-H2/38-H, the multiplicity and even chemical shifts reported in table 1 are not likely. The geminal coupling of the 37-H2 protons should correspond to a J value of at least 14 Hz, as reported for voluhemins featuring the same epoxymethilene moiety linked to the aromatic ring. And this J value is clearly evident from the CDCl3 spectrum given in the supporting material. I strongly recommend the authors to review all the assignments in DMSO-d6 reported in table 1 and, if possible, where the correct assignment of the chemical shifts and / or the multiplicity of some resonances is not allowed due to the overlapping of the signals, to add the assignments made in CDCl3.
- Other minor corrections:
Pag. 2, line 46. “In the present study, we describe…”, not described
Pag. 2, line 47. “…fermentation…of these compounds” is not correct, please remove fermentation.
Pag. 3, line 67. Figure 60??? May be Fraction 60%-2? Please correct.
Pag. 4, line 109. “…trisubstituted benzene ring” not ““…trisubstituted aromatic ring”.
Pag. 7, line 192. “…the 1H-NMr spectrum of 3”, not “...of 1”
Pag. 8, line 214. Remove “that” after moiety
Reviewer 2 Report
The manuscript molecules-853052 reports interesting new data on terpendoles from the fungus Volutella citrinella.
The manuscript is well organized, the methodological approach is correct and adequate to the research. The discussion needs to be implemented, but I consider that this manuscript is appropriate for publication in Molecules with only minor revisions.
The aim of the research can be understood, but it should be better explained in the introduction and the authors could anticipate it in the abstract, given that it is the latter is short enough.
Scheme 1 should be referred to as Figure 2.
Lines 56-63. This is a description of the methodological approach and therefore should be included in chapter 4.
Lines 67-69. "Figure 60. (brown material, 249 mg) was subjected to preparative high-performance liquid ...". There is something wrong and in any case, this would also go to chapter 4.
Figures S3-S9, S11-S13, S15-S16, S18, S21-22 and S24 are not mentioned in the text.
The discussion should be implemented, in particular in relation to SOAT activity inhibition which is a very interesting issue.
Reviewer 3 Report
The manuscript by Elyza Aiman Azizah Nur and co-workers described isolation and structural determination of new terpendole congeners, terpendoles N-P, from Volutella citronella BF-0440. Their structures were determined through extensive NMR analysis. Structure-activity relationship (SAR) of terpendoles as sterol O-acyltransferase 1 and 2 isozymes were also described.
The work is scientifically sound and potentially interesting to the researchers in the fields of natural product chemistry and chemical biology. Especially, the SAR analysis reported here provides opportunities to develop a post-statin drug. Overall, the reviewer thinks that this work includes quality required for publication in this journal after the authors have addressed the minor points as follows.
Minor comments
1. Page 3, line 67, Figure 60. was subjected to …
Figure 60 ?
2. Page 3, line 68, high-performance liquid chromatography (HPLC; column…)
HPLC conditions described in the main text should be summarized in “Materials and Methods” section.
3. Page 3, line 113, from 41-H3 (d 24.6) to C-38, C-39, and C-41…
C-41 should be C-40.
4. Page 5, table 2, 1H-NMR analysis.
Based on the structure of 2, multiplicities of H37 {2.93 (tt, 6.0, 2.0) and 3.01 (t, 7.6, 8.4)} and H38 {2.93 (tt, 6.0, 2.0)} are likely strange. In my opinion, H38, which is coupled to H37a and H37b, gives a dd instead of a tt. This mistake might be due to the overlapping of H37 and H38 signals. Please also check the following signals.
Compound 3
H20; 7.26 (t, 7.8):: triplet? It looks like a doublet.
H23; 7.24 (t, 7.8):: triplet? It looks like a doublet.
H33; 5.18 (tt, 7.0, 1.2):: tt?? It looks like a doublet of doublets.
5. Page 7, Figure 3 (a)
This figure is difficult to follow because many arrows around C25-Me are overlapped. This reviewer recommends separating the indole ring from the B-ring to make some space around C25-Me.
6. Page 7, Figure 4 (b)
In my opinion, the authors should focus on key correlations for elucidating the relative stereochemistry. According to this line, this reviewer recommends to delete the correlations between (H9 and H29) and (10-OH and H28) because they gave no information about the relative stereochemistry.
Reviewer 4 Report
The manuscript regarding the isolation of new terpendoles is very interesting. The authors presented in detail the process of compound isolation and spectral characteristics of new derivatives.
I have only a few comments.
In line 67 there is the term Figure 60, which is incorrectly used here
Lines 71-71, CH3CN solution was 80% or 78%
Why in Table 2 the order of compounds is as follows: 2, 1, 3
Included in Table 2 signals (compound 2) coming from groups CH2-37 and CH-38 raise doubts, in both cases there is 2.93 (tt, 6.0, 2.0)
Reviewer 5 Report
This is a very good paper on extraction, purification, structural elucidation and discussion on structure-activity relationship of a series of terpendoles, a fungal metabolites.
The argument of the paper is of importance to the field, the used experimental design and the technical approach are of good quality. The manuscript is well written, and the data are correctly presented and discussed by the Authors.
Probably from line 63 to line 67 some sentence is missing. Please correct.
In my opinion the manuscript can be accepted for publication without any substantial modification.
